# Learning to See the Unseen: Few-Shot 3D Scene Reconstruction via Diffusion and Gaussian Fields

## Abstract

While recent methods have achieved impressive results in 3D reconstruction, they typically rely on dense multi-view inputs and often struggle with ambiguity in occluded or unobserved regions, particularly in complex scene layouts and background areas. We propose Learning to See the Unseen (LSU), a unified framework for high-fidelity 3D scene reconstruction from sparse or even single-image inputs by coupling generative novel-view synthesis with Gaussian-based scene reconstruction. Our approach introduces a Scene Diffusion Module (SDM) that conditions on sparse views and text prompts to synthesize consistent novel views. To improve spatial alignment across generated views, SDM incorporates a scene-level geometric supervision strategy that constrains the diffusion process using 3D structural consistency. Additionally, we design a geometry-aware Gaussian reconstruction module that leverages depth and surface normal priors to refine the reconstructed scene, improving geometric accuracy, background coherence, and rendering fidelity. Extensive experiments demonstrate that LSU achieves state-of-the-art performance on the RealEstate10K dataset and generalizes effectively to unseen domains, including KITTI and Mip-NeRF, recovering accurate global geometry while preserving fine-grained visual details across diverse scenes.

## 1 Introduction

3D reconstruction plays a crucial role in enabling realistic scene understanding, novel view synthesis, and immersive content creation. While object-centric methods have achieved impressive performance, they struggle to generalize to unbounded scenes where occlusion, asymmetry, and missing viewpoints are common. These challenges become particularly severe in single or sparse-view reconstruction settings, where geometric cues and multi-view constraints are inherently limited.

Neural Radiance Fields (NeRFs) (Mildenhall et al., 2021) and Gaussian Splatting (Kerbl et al., 2023) have demonstrated remarkable success in photorealistic 3D reconstruction and novel view synthesis. However, their success heavily depends on dense multiview supervision with well-distributed viewpoints. When viewpoints are sparse, NeRF-based methods often produce incomplete geometry, oversmoothed textures, and view-dependent artifacts (Martin-Brualla et al., 2021). Although recent efforts like SparseNeuS (Long et al., 2022) aim to mitigate these drawbacks, quality degradation is still noticeable as viewpoint coverage decreases. Similarly, Gaussian splatting requires multiple views to accurately estimate Gaussian primitives, especially in complex scenes (Xu et al., 2024).

Recent methods have explored 3D reconstruction using fewer images, reducing the need for dense views. Methods such as pixelSplat (Charatan et al., 2024), latentSplat (Wewer et al., 2024), and MVSplat (Chen et al., 2024) reconstruct 3D from only two input views by leveraging Gaussian splatting and interpolation techniques. However, these approaches struggle with occlusion and often fail to recover geometry for regions not directly observed. Beyond sparse-view approaches, single-image reconstruction approaches attempt to infer a complete 3D structure from a single viewpoint. Methods such as Behind the Scenes (BTS) (Wimbauer et al., 2023), MINE (Li et al., 2021), and Flash3D (Szymanowicz et al., 2025) rely heavily

---

[0]From here onwards, "sparse/few shots" will be used interchangeably.

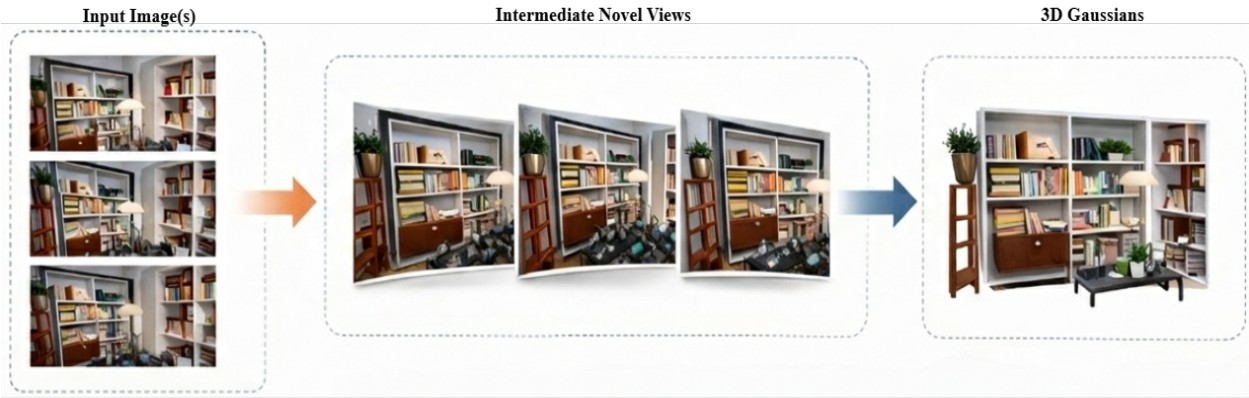

Figure 1: **Model Overview:** Given one or more sparse input views, the model generates multi-view images and transforms them into a 3D Gaussian representation.

on depth-based representations, including implicit depths, multiplane depths, or depth-guided Gaussian primitives. Wonderland (Liang et al., 2025) further pushes this direction by deriving 3D Gaussians from video latents generated from a single input image. However, its reliance on a single-view input restricts its applicability in sparse-view settings.

While previous methods focus on either single-view or sparse-view 3D reconstruction, a natural gap remains in developing a unified framework that can effectively handle both settings. Moreover, although the diffusion objective enforces denoising consistency in the latent space, it does not explicitly constrain the geometric accuracy or spatial alignment of the reconstructed 3D scene. Motivated by these limitations, we introduce LSU (Learning to See the Unseen), a unified framework for 3D reconstruction from sparse or even single-image inputs that combines generative novel-view synthesis with a geometry-aware Gaussian reconstruction pipeline. LSU incorporates explicit scene-level geometric supervision during novel-view synthesis, encouraging the generated views to remain aligned with the underlying scene structure. In the reconstruction stage, LSU further leverages depth and surface normal priors to refine the Gaussian representation, leading to more accurate geometry, sharper appearance, and improved structural fidelity. As a result, LSU enables realistic 3D scene reconstruction even when regions are occluded.

- We propose LSU, a unified framework for 3D scene reconstruction from sparse or even single-image inputs by coupling generative novel-view synthesis with Gaussian-based scene reconstruction.

- We introduce the Scene Diffusion Module (SDM), which conditions on sparse views and text prompts, and incorporates a scene-level geometric supervision strategy for novel-view synthesis that constrains the diffusion model using 3D structural consistency, improving spatial alignment across generated views.

- We design a geometry-aware Gaussian reconstruction module that incorporates depth and surface normal priors to refine the reconstructed scene and improve rendering fidelity.

- Extensive experiments show that LSU achieves state-of-the-art performance on RealEstate10K (Zhou et al., 2018) and generalizes effectively to unseen domains such as KITTI (Geiger et al., 2012) and Mip-NeRF (Barron et al., 2022).

## 2 Related Work

### 2.1 3D Scene Reconstruction

Although 3D object reconstruction has seen substantial progress (Yu et al., 2021; Liu et al., 2023; Chan et al., 2023), large-scale scene reconstruction remains challenging, especially under sparse-view settings.

Scene-level generation must capture wider spatial extents, complex layouts, and severe occlusions, which limits the effectiveness of object-centric approaches. NeRF-based methods such as Mip-NeRF 360 (Barron et al., 2022), Zip-NeRF (Barron et al., 2023), and SparseNeuS (Long et al., 2022) improve scalability and surface quality, but they still require dense multiview coverage. Gaussian Splatting approaches like 3D-GS (Kerbl et al., 2023) offer efficient rendering with Gaussian primitives, yet they also depend on extensive observations for stable geometry and appearance.

To mitigate data requirements, lifting-based techniques such as pixelSplat (Charatan et al., 2024), latentSplat (Wewer et al., 2024), and MVSplat (Chen et al., 2024) reconstruct 3D scenes from as few as two input images. These methods estimate Gaussian primitives directly from image pairs, achieving visually coherent results in observed regions. However, they are primarily interpolative, limited to visible surfaces, and unable to infer geometry or texture in unseen parts of the scene.

In summary, existing approaches are largely constrained by their input coverage: dense-view methods require many images, sparse-view methods fail to infer occluded geometry, and single-image methods struggle with large viewpoint changes. In contrast, our framework leverages video diffusion models for generative modeling of unseen and occluded regions, producing geometrically complete and visually consistent 3D reconstructions even from sparse or single-image inputs.

## 2.2 Video Diffusion

Diffusion Models (DMs) have achieved strong performance in image (Rombach et al., 2022) and video generation (Ho et al., 2022). Latent Diffusion Models (LDMs) (Rombach et al., 2022) operate in compressed latent spaces for efficiency, and Video Diffusion Models (VDMs) extend this framework to generate temporally coherent videos. However, using VDMs for view-consistent 3D generation remains underexplored, as most methods prioritize temporal smoothness rather than cross-view spatial alignment.

For novel view synthesis, ZeroNVS (Sargent et al., 2023) and Zero-1-to-3 (Liu et al., 2023) can generate unseen viewpoints but often suffer from inaccurate poses and loss of geometric detail. ViewCrafter (Yu et al., 2024) improves fidelity using sparse point clouds, yet still struggles with consistent 3D alignment. Cat3D (Gao et al., 2024b) and ReconFusion (Wu et al., 2024) further explore diffusion-based 3D generation from limited observations, but often suffer from cross-view inconsistency and unstable scene geometry. Wonderland (Liang et al., 2025) reconstructs Gaussian scenes from video latents produced from a single image, but it lacks explicit novel-view generation and often misses fine details.

In contrast, our approach enforces explicit spatial alignment between generated views and reconstructed geometry through a dedicated 3D scene loss. LSU conditions on sparse image inputs and optional text prompts to produce semantically and geometrically consistent novel views, which are further refined with depth and surface normal priors. Unlike single-image methods such as Wonderland, LSU generalizes effectively to sparse-view inputs and maintains strong geometric consistency across views.

# 3 Methodology

This section presents the overall framework of our approach. We begin by outlining the fundamental concepts underlying video diffusion models, which form the foundation of our scene-level synthesis pipeline. We then introduce the proposed Scene Video Diffusion Model (SDM) for consistent novel-view generation, followed by the Gaussian Reconstruction Module, which translates synthesized views into coherent 3D representations.

## 3.1 Preliminary: Video Diffusion Models

Diffusion models (Ho et al., 2020; Sohl-Dickstein et al., 2015) are generative models that learn to denoise by eliminating noise from clean data. The forward process $q(x_t|x_0, t)$ transforms the data $x_0 \sim p(x)$ into Gaussian noise $x_T \sim \mathcal{N}(0, I)$ over $T$ time steps by gradually adding noise to $x_0$ to yield $x_t$, where $x_t$ is given by the equation $x_t = \alpha_t x_0 + \sigma_t \epsilon$, with $\epsilon \sim \mathcal{N}(0, I)$. The reverse process $p_\theta(x_{t-1} \mid x_t, t)$ emphasizes the use of a noise predictor $\epsilon_\theta(x_t, t)$ to eliminate noise from clean data, which is supervised by the objective:

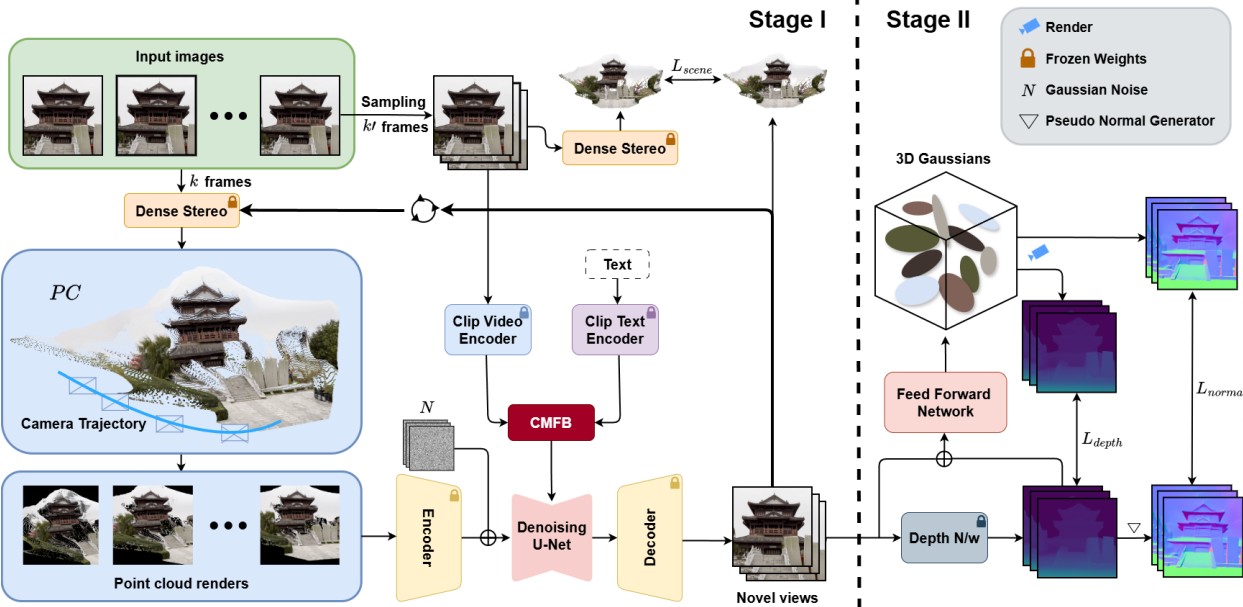

Figure 2: **Overview of LSU.** A sparse point cloud is first generated from the selected $k$ input frames and rendered to produce the conditioning views to the diffusion module. The sampled $k'$ video frames, along with optional text prompts, are then encoded using the Cross-Modal Fusion Block (CMFB) to guide novel view synthesis. The generated views are lifted into 3D Gaussian representations and refined using depth and normal supervision to ensure a consistent and coherent reconstruction.

$$\min_\theta \mathbb{E}_{t,\, x \sim p(x),\, \epsilon \sim \mathcal{N}(0,I)} \big\| \epsilon - \epsilon_\theta(x_t, t) \big\|_2^2 \tag{1}$$

Building on this formulation, we present a two-stage 3D scene reconstruction pipeline, as illustrated in Fig. 2. The first stage focuses on synthesizing novel, view-consistent perspectives using Latent Diffusion Models (LDMs), which are widely utilized in video generation tasks and naturally align with our goal of ensuring cross-view consistency. The second stage employs a Gaussian-based reconstruction module that leverages these synthesized views to produce a spatially aligned 3D representation.

### 3.2 Novel View Synthesis with Scene Video Diffusion Model

A key factor in successful 3D reconstruction is the availability of a large number of input views; however, in single-view or sparse-view settings, this becomes a limiting factor, motivating the need to synthesize additional views to improve reconstruction quality. The straightforward option is to use a basic diffusion model or a depth-based method. Although both techniques are effective for object-level view synthesis, we observe that they face significant challenges when applied to large-scale scenes, often resulting in views that are inconsistent with substantial occluded regions. To address this problem, we propose a scene-level view-synthesis diffusion module.

Given a small set of input views $\mathcal{V} = \{\mathbf{V}_i\}_{i=1}^N$ and a text description $t$, we first reconstruct a sparse point cloud **PC** using the dense stereo model (DSM) (Wang et al., 2024). The DSM operates on a subset of $k$ views from $\mathcal{V}$ to generate a coarse geometric prior of the scene. The reconstructed **PC** is then projected onto the image plane along the camera trajectory to generate a sequence of rendered images, which are encoded by $\mathcal{E}$ into a latent representation $z_t$. These latent features preserve visible structures while indicating missing or occluded regions. The representation is then combined with Gaussian noise and passed to the diffusion module for denoising-based synthesis.

The remaining views $\mathcal{V}_{N-k}$ are used as candidate conditioning inputs from which we sample up to $k'$ views, typically $k' = 4$, to provide complementary context. During training, we select two conditioning views from camera poses within the convex hull of the input cameras to encourage interpolation, and two additional views from nearby poses slightly outside the input range to promote extrapolation. To support a variable number of conditioning inputs, we apply conditioning dropout by replacing each conditioning view with a learned null embedding with a fixed probability, and for single-view inputs, only the given image is kept non-null while the remaining slots are filled with null embeddings. Finally, the sampled conditioning views and the text description $t$ are encoded to obtain $\mathbf{F}_v$ and $\mathbf{F}_t$, which condition the diffusion process.

To establish a relationship between the visual and textual modalities, thereby enabling the model to capture the scene's underlying geometry and temporal dynamics, we introduce a Cross-Modal Fusion Block (CMFB). The CMFB enables bidirectional interaction between the visual and textual representations, allowing semantic cues from the text to guide the interpretation of visual–temporal patterns in the video features. The cross-modal integration enhances the model's ability to align semantic understanding with geometric structure, resulting in more coherent and realistic novel views.

The diffusion model is trained to learn a conditional distribution that can be reversed to synthesize novel, view-consistent renderings. Specifically, the model is optimized to predict the noise added during the forward diffusion process using a denoising objective, formulated as:

$$\mathcal{L}_{\text{diff}} = \mathbb{E}_{\mathcal{E}(x),\,\epsilon\sim\mathcal{N}(0,I),t}\big\|\epsilon - \epsilon_\theta(z_t; y, t)\big\|_2^2 \tag{2}$$

where $y$ serves as the conditioning signal and $z_t$ is obtained by passing $x_t$ through the encoder $\mathcal{E}$. The generated views from the decoder $\mathcal{D}$ are used to iteratively update the point cloud $\mathbf{PC}$; the refinement is carried out a fixed number of times (say, 10 iterations), where in each iteration a subset of novel views is sampled and passed through the dense stereo model, resulting in progressively denser and less noisy reconstructions over time, creating better novel views.

### 3.3 Scene weighted Loss

While the diffusion objective enforces denoising consistency in the latent space, it does not explicitly constrain the geometric accuracy or spatial alignment of the reconstructed 3D scene. To address this limitation, we introduce a scene-level loss computed directly in 3D space. Specifically, we compare two point clouds: a reference point cloud $\mathcal{P}$ obtained by passing selected input views through a dense stereo model, and a reconstructed point cloud $\mathcal{Q}$ obtained by projecting the predicted depth maps using (Yang et al., 2024) from the generated novel views. This point-level supervision encourages geometric consistency between the reconstructed and reference scenes, ensuring that improvements in the latent representation translate into physically plausible spatial structures.

Since the geometry estimated by the dense stereo model may contain noisy predictions, we incorporate confidence-aware supervision. Let $c_i \in [0,1]$ denote the confidence score associated with each point $p_i \in \mathcal{P}$. We first apply confidence-based filtering to discard unreliable points:

$$\mathcal{P}_f = \{p_i \in \mathcal{P} \mid c_i > \tau\}, \tag{3}$$

where $\tau$ is a confidence threshold. The remaining points are then assigned normalized weights

$$w_i = \frac{c_i}{\sum_{p_j \in \mathcal{P}_f} c_j}, \tag{4}$$

so that higher-confidence points exert stronger supervision during training.

The final scene loss is formulated as a confidence-weighted symmetric distance between the filtered reference point cloud $\mathcal{P}_f$ and the reconstructed point cloud $\mathcal{Q}$:

$$\mathcal{L}_{\text{scene}} = \sum_{p_i \in \mathcal{P}_f} w_i \min_{q \in \mathcal{Q}} \|p_i - q\|_2^2 + \frac{1}{|\mathcal{Q}|} \sum_{q \in \mathcal{Q}} \min_{p_i \in \mathcal{P}_f} \|q - p_i\|_2^2. \tag{5}$$

For stable training, we apply this supervision only at denoising steps corresponding to low-noise forward-process timesteps, where the corrupted input $x_t$ remains close to the clean image. In this regime, the model's reconstructions exhibit higher visual fidelity, providing more reliable geometric signals for supervision.

The final objective combines both the latent-space diffusion loss and the proposed scene-level geometric constraint:

$$\mathcal{L} = \lambda_{\text{diff}}\mathcal{L}_{\text{diff}} + \lambda_{\text{scene}}\mathcal{L}_{\text{scene}}, \tag{6}$$

where $\mathcal{L}_{\text{diff}}$ operates in the latent image space, promoting perceptually coherent and view-consistent image synthesis, while $\mathcal{L}_{\text{scene}}$ enforces accurate spatial correspondence and structural integrity in the reconstructed 3D geometry.

In our implementation, we adopt DUSt3R to obtain a reference point cloud for geometric supervision due to its robustness in sparse-view settings. Importantly, the proposed scene-level supervision is agnostic to the reconstruction backbone and requires only a differentiable mechanism for estimating point cloud structures. Note that the dense stereo model is used solely to provide geometric supervision during training and is not required during inference.

### 3.4 Gaussian Reconstruction

While the diffusion-based module provides globally consistent novel views, accurate scene-level reconstruction requires explicit modeling of fine-grained geometry, particularly in regions that are sparsely observed or heavily occluded. To achieve this, we develop a Gaussian reconstruction framework that incorporates geometric priors such as depth and surface normals to enhance local structural fidelity and spatial smoothness.

The reconstructed Gaussian representation is parameterized by attributes including color, opacity, position, and scale. Depth and normal priors serve as additional constraints, ensuring that the reconstructed surfaces adhere to physically consistent geometry. Building upon the synthesized novel views produced by the diffusion model, we employ a feed-forward network that predicts Gaussian parameters anchored by the estimated depth, enabling efficient and generalizable reconstruction across diverse scenes. This unified approach not only provides efficient and generalizable Gaussian reconstruction but also ensures consistency across diverse scenes without requiring per-scene fine-tuning.

To ensure accurate geometric supervision, we define multiple complementary loss functions.

**Depth loss:** We use a depth prior $D$ predicted from the pre-trained model (Yang et al., 2024) to supervise the rendered depth $\hat{D}$ :

$$\hat{D} = \sum_{i \in \mathcal{N}} d_i \alpha_i \prod_{j=1}^{i-1}(1 - \alpha_j) \tag{7}$$

$$\mathcal{L}_{\text{depth}} = \frac{1}{|P|}\sum \left[ \exp\left(-|\nabla I|\right) \cdot \left( \left\|\hat{D} - D\right\|_1 \right) \right]$$

where $d_i$ is the $i$-th Gaussian z-depth coordinate in view space, $\alpha_i$ is the blending coefficient for a Gaussian, $P$ is the set of valid pixels, and $\nabla I$ is the gradient of the rgb image. The scale ambiguity of the Depth Anything v2 (Yang et al., 2024) is resolved using the scale-alignment strategy of (Chung et al., 2024). Unlike the original work, which aligns predicted depth to sparse SfM points, we align the Depth Anything v2 predictions to the rendered depth from our Gaussians by optimizing a global scale and shift.

**Smooth Normal Loss**: Estimating the normal maps using a pre-trained model introduces scale discrepancies and inconsistencies with the depth map across various regions, which can negatively impact model performance; to address this, we use pseudo-ground truth normals $\bar{N}_d$ calculated from the depth map by first lifting depth values into 3D space using camera intrinsics and then computing surface normals via spatial derivatives of the resulting 3D points, following (Gao et al., 2024a). These pseudo-normals are used to calculate the normal loss using L1 and cosine similarity terms:

$$\mathcal{L}_{\text{normal}} = \left\|\bar{N}_d - \hat{N}\right\|_1 + \left(1 - \bar{N}_d \cdot \hat{N}\right) \tag{8}$$

| Input | Target | MINE-64 | Flash3D | Ours |
| --- | --- | --- | --- | --- |

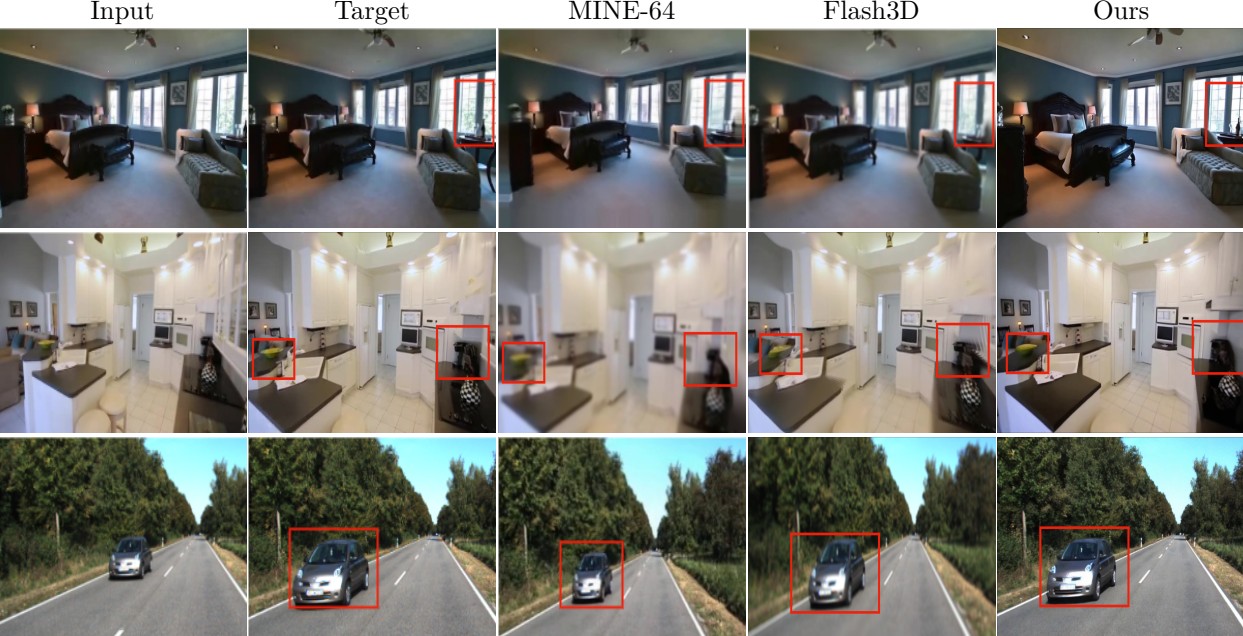

Figure 3: Qualitative comparison of prior methods across diverse datasets. The first two rows show results on the in-domain RealEstate10K dataset, while the last row presents results on the cross-dataset KITTI. LSU consistently produces sharper views compared to prior methods. Red bounding boxes highlight regions of interest to assist visual comparison.

where $\hat{N}$ represents the rendered normal from the Gaussian model.

The overall Gaussian optimization loss combines RGB, depth, and normal supervision terms as:

$$\mathcal{L}_{\text{gaussian}} = \mathcal{L}_{\text{rgb}} + \lambda_{depth}\mathcal{L}_{\text{depth}} + \lambda_{normal}\mathcal{L}_{\text{normal}} \tag{9}$$

where $\mathcal{L}_{\text{rgb}}$ is the orginal photometric loss proposed in Gaussian Splatting (Kerbl et al., 2023). See the supplement for more details.

## 4 Experiments

We present our experimental investigation to show the effectiveness of LSU in both in-domain settings and cross-dataset generalisation for single-view 3D reconstruction. Second, we perform a sparse-view comparison to assess the model's performance under a limited-view setting. Finally, we show via ablation studies how each design choice contributes to the performance. Our model is trained on a combination of large-scale datasets, including WebVid-10M (Bain et al., 2021), RealEstate10K (Zhou et al., 2018), and DL3DV-10K (Ling et al., 2024).

### 4.1 3D Scene Generation from Single Image

We evaluate the quality of reconstruction by training and testing on the RealEstate10K (Zhou et al., 2018) dataset. Since the dataset only contains RGB images of a scene, we evaluate the reconstruction quality by rendering novel views from the reconstructed scene.

**Metrics.** For quantitative evaluation, we assess both visual quality and temporal coherence. Visual quality is measured using standard image-based metrics, including PSNR, SSIM (Wang et al., 2004), and LPIPS (Zhang et al., 2018). In addition, the Fréchet Inception Distance (FID) (Heusel et al., 2017) is used to jointly evaluate overall visual fidelity and temporal coherence across generated sequences.

| Model | In-domain | | | Cross-dataset | | |
|---|---|---|---|---|---|---|
| | RealEstate10K | | | KITTI | | |
| | PSNR ↑ | SSIM ↑ | LPIPS ↓ | PSNR ↑ | SSIM ↑ | LPIPS ↓ |
| SV-MPI | 23.52 | 0.78 | - | 19.50 | 0.73 | - |
| BTS | 24.00 | 0.75 | 0.19 | 20.10 | 0.76 | - |
| MINE | 24.75 | 0.82 | 0.17 | 21.90 | 0.82 | 0.11 |
| Flash3D | 24.93 | 0.83 | 0.16 | 21.96 | 0.82 | 0.13 |
| LSU (Ours) | **25.34** | **0.86** | **0.15** | **21.98** | **0.83** | **0.11** |

Table 1: **Quantitative comparison on 3D scene renderings in single target view setting against regression-based methods.** We outperform all baseline methods in terms of PSNR, SSIM, and LPIPS on the real-world RealEstate10K dataset (in-domain). For the cross-dataset setting, our method achieves better results on the KITTI dataset.

| Model | PSNR ↑ | SSIM ↑ | LPIPS ↓ |
|---|---|---|---|
| ZeroNVS | 13.01 | 0.37 | 0.44 |
| ViewCrafter | 16.84 | 0.51 | 0.34 |
| Wonderland | 17.15 | 0.55 | 0.29 |
| LSU (Ours) | **21.56** | **0.75** | **0.22** |

Table 2: **Quantitative comparison on 3D scene renderings in multi-target view setting against generative methods.** We outperform all baseline methods in terms of PSNR, SSIM, and LPIPS on the real-world RealEstate10K dataset. The evaluation is performed on 2D views rendered from our 3D Gaussians.

**Baselines.** We compare our method with Flash3D (Szymanowicz et al., 2025), MINE (Li et al., 2021), SV-MPI (Tucker & Snavely, 2020), BTS (Wimbauer et al., 2023) and Wonderland (Liang et al., 2025). Following the evaluation protocol used in prior works such as Flash3D, MINE, BTS, and SV-MPI, we assess rendering quality by randomly selecting a target view within a 30-frame window and comparing the rendered result from the 3D scene to the ground-truth view.

For a fair comparison with Wonderland (Liang et al., 2025), ViewCrafter (Yu et al., 2024), we use two additional evaluation settings: 3D scene comparison and camera-guided setting for novel view synthesis.

In the 3D scene generation, we sample 100 test images from RE10K and generate the 3D scene using the single input image. From the 3D scene, 14 frames are rendered and are evaluated against temporally aligned frames from the corresponding ground-truth videos. In the camera-guided setting, we randomly select 300 test images with their associated RE10K videos. For each video, the first frame serves as the input, and the remaining camera poses are provided as conditioning during generation. We then compare the first 14 generated frames to the ground-truth frames with matching poses.

**Discussion.** As shown in Tables 1, 2, and 3, our method achieves state-of-the-art performance on PSNR, SSIM, and LPIPS compared to previous works. In addition, Tab. 3 reports results on FID, where our method also achieves superior performance, indicating better overall visual quality and temporal coherence. Qualitative results in Fig. 3 further demonstrate the effectiveness of our approach. Renderings produced by MINE (Li et al., 2021) and Flash3D (Szymanowicz et al., 2025) often appear distorted in certain regions and noticeably blurry, whereas our method leverages the generative capabilities of the model to better handle occluded regions and produce sharper, more consistent views.

## 4.2 Cross-Dataset Reconstruction

To discuss the cross-dataset generalization ability of our model, we conduct experiments by testing it on the unseen outdoor KITTI dataset (Geiger et al., 2012). Table 1 shows that our model outperforms Flash3D

| Method | FID ↓ | PSNR ↑ | SSIM ↑ | LPIPS ↓ |
|---|---|---|---|---|
| ViewCrafter | 20.89 | 18.91 | 0.50 | 0.21 |
| Wonderland | 16.16 | 19.71 | 0.55 | 0.20 |
| LSU (Ours) | **14.75** | **23.10** | **0.82** | **0.18** |

Table 3: **Quantitative comparison of camera-guided setting** We report FID, PSNR, SSIM, and LPIPS scores, showing that our method achieves the best overall performance.

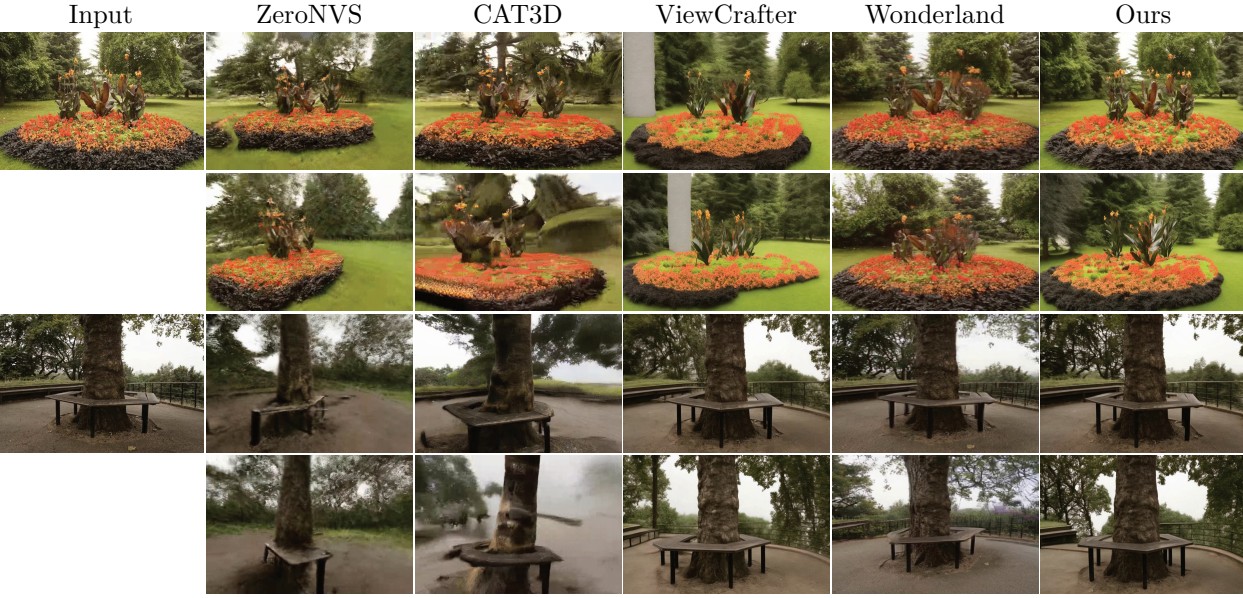

Figure 4: Qualitative comparison on the Mip-NeRF dataset with a single view. For each scene, the left column shows the input (conditional) image, along with renderings from two viewpoints: the upper image corresponds to the view closest to the input (conditional) view, while the lower image shows a view approximately 70° rotated from it.

(Szymanowicz et al., 2025) on the KITTI dataset. This highlights our model's ability to generalize across diverse environments.

As shown in Fig. 4, we demonstrate the effectiveness of our method in single-image reconstruction by comparing it with other diffusion-based approaches for Mip-NeRF (Barron et al., 2022). The first row shows a view close to the input, while the second depicts a view 70° from the input view. Due to the lack of publicly available code for CAT3D and Wonderland, we sourced their comparison results from their official webpages (Gao et al., 2024b; Liang et al., 2025) and aligned our camera trajectories accordingly. The results indicate that LSU generates sharper imagery and exhibits greater consistency in geometry and appearance along shorter trajectory ranges, relative to the input view, compared to other methods, even though it is not trained on that dataset. In contrast, ViewCrafter (Yu et al., 2024) produces noticeable artifacts in both near-input and distant viewpoints, which diminishes its overall robustness. Furthermore, LSU produces substantially clearer foreground and performs on par with Wonderland in rendering background content over longer trajectory ranges.

## 4.3 Few-Shot Reconstruction

**Baselines.** We compare our method with pixelSplat (Charatan et al., 2024), latentSplat (Wewer et al., 2024), MVSplat (Chen et al., 2024), MVGD (Guizilini et al., 2025), Cat3D (Gao et al., 2024b), ReconFusion (Wu et al., 2024). Since the official implementations of MVGD, Cat3D, and ReconFusion are not publicly

Input Views      MVSplat      latentSplat      Ours

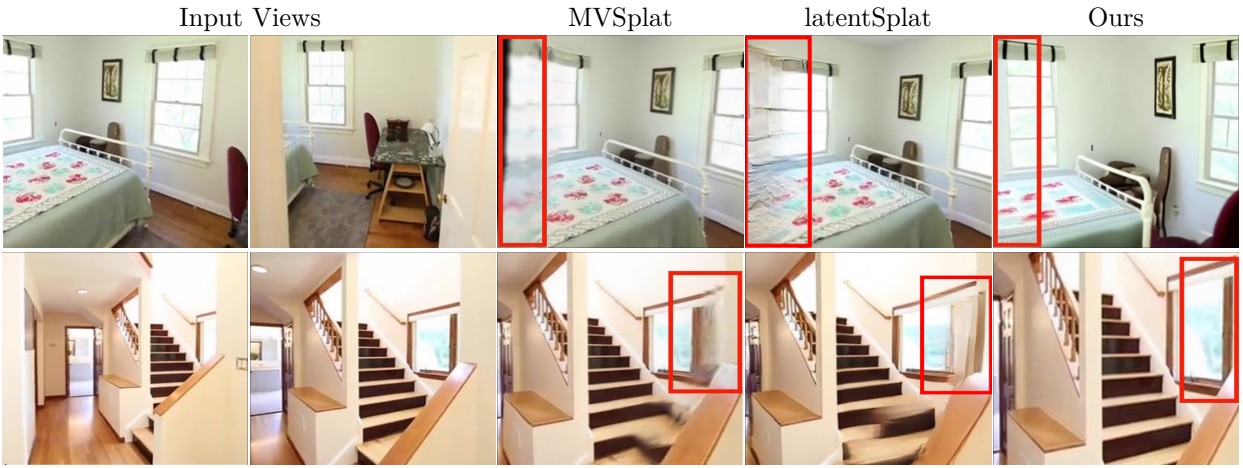

Figure 5: Qualitative comparison on RealEstate10K with 2 input images.

Input      ReconFusion      CAT3D      Ours

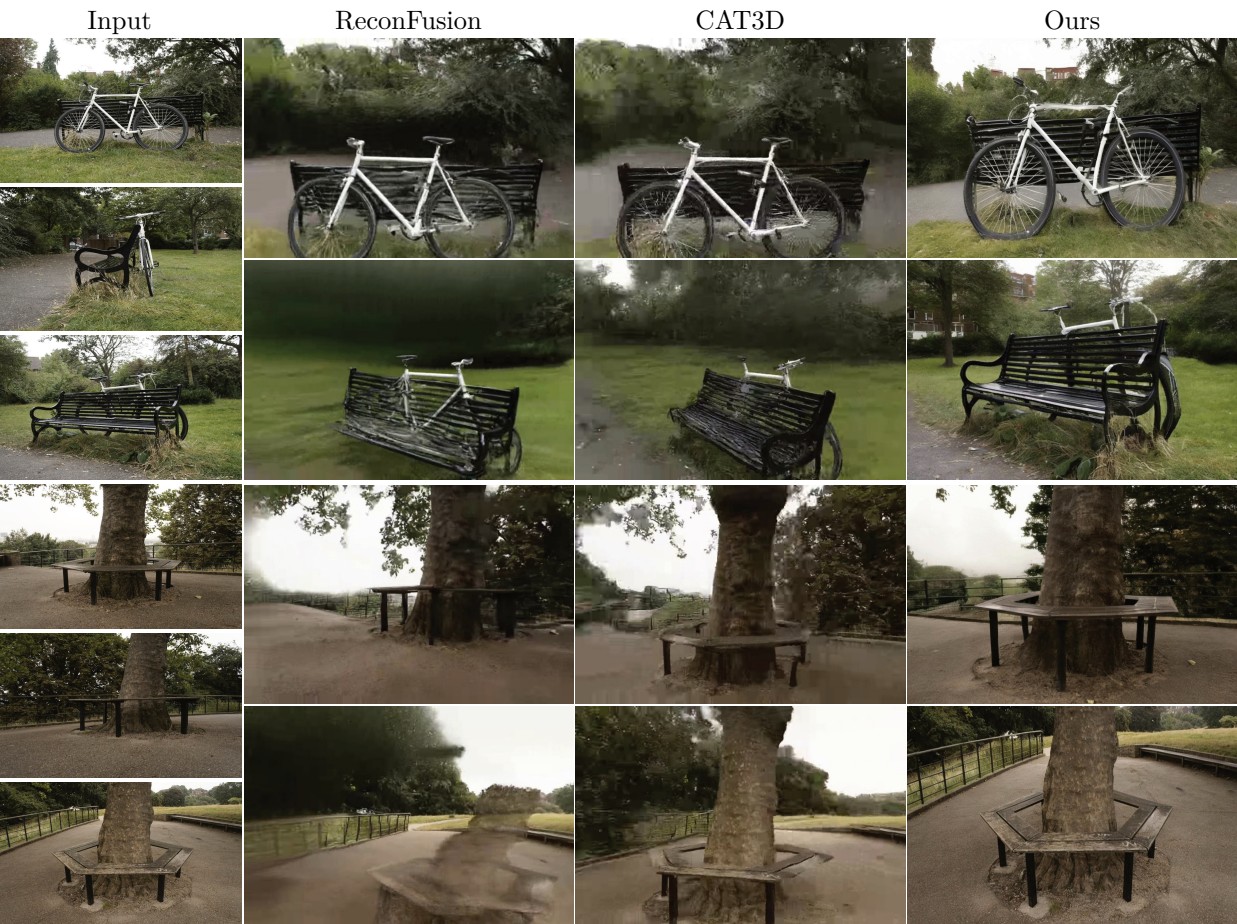

Figure 6: Qualitative comparison on the Mip-NeRF dataset with 3 input views, where the first column shows compressed input views and the remaining columns show reconstructed outputs.

available, we follow the evaluation protocol described in these works, which reports results on a single target frame. However, we extend this setting to use three randomly selected target frames for a more comprehensive and fair comparison.

| Method | Views | RE10k Interpolation | | |
|---|---|---|---|---|
| | | PSNR ↑ | SSIM ↑ | LPIPS ↓ |
| pixelSplat | 2 | 26.09 | 0.86 | 0.13 |
| latentSplat | 2 | 23.93 | 0.81 | 0.16 |
| MVSplat | 2 | 26.39 | 0.86 | 0.12 |
| MVGD | 2 | 28.41 | 0.89 | 0.10 |
| ReconFusion | 3 | 25.84 | **0.91** | 0.14 |
| CAT3D | 3 | 26.78 | **0.91** | 0.13 |
| LSU (Ours) | 2 | **28.61** | **0.91** | **0.09** |

Table 4: Comparison on RE10k Interpolation with different methods using sparse input views. The best and second best are marked in **bold** and underline.

**Discussion.** As reported in Tab. 4, LSU consistently outperforms existing two-view synthesis methods in terms of interpolation quality between two views. Remarkably, our two-view synthesis results even surpass several recent three-view synthesis approaches, demonstrating the strong generalization capability of our method. Furthermore, as illustrated in Fig. 5, LSU produces sharper details and more faithful structures compared to competing approaches, highlighting its superior ability to preserve scene geometry and appearance. Figure 6 presents a qualitative comparison between our method and existing three-view scene reconstruction approaches on the Mip-NeRF dataset. These results affirm that our model can generate high-fidelity, geometrically consistent 3D scenes in sparse-view scenarios.

| RGB | Depth | Normal | PSNR ↑ |
|---|---|---|---|
| ✓ | ✗ | ✗ | 21.02 |
| ✓ | ✓ | ✗ | 21.50 |
| ✓ | ✓ | ✓ | 21.56 |

Table 5: Ablation study on 3D reconstruction for RealEstate10K.

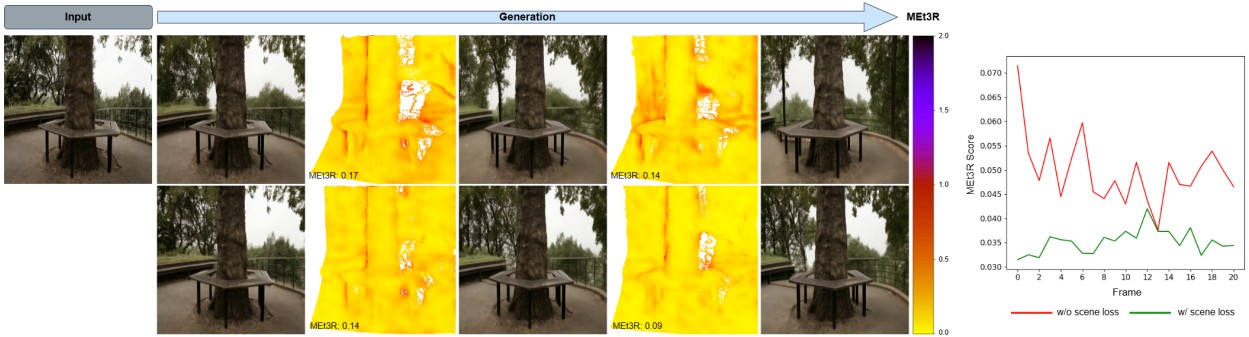

Figure 7: **Left:** Generated images without scene loss (top row) and with scene loss (bottom row), conditioned on the input frame. The MEt3R score maps indicate the level of inconsistencies between frame $i$ and frame $i+5$. **Right:** Pairwise consistency scores evaluated for consecutive frames using a sliding window.

### 4.4 Ablation

We analyze the impact of geometric supervision in the Gaussian reconstruction stage on RealEstate10K Zhou et al. (2018), with results summarized in Tab. 5. Specifically, we ablate the depth and normal objectives by removing each loss term individually while keeping all other settings fixed. As shown in Tab. 5, removing the depth loss leads to the largest degradation, reducing PSNR to 21.02, which indicates that depth supervision

provides a critical geometric signal for accurate reconstruction. Removing the normal loss yields a smaller but consistent drop to 21.50, suggesting that normal supervision further improves surface consistency. The full model achieves the best performance with a PSNR of 21.56.

To evaluate view consistency between frames, we employ MEt3R (Asim et al., 2025), and we refer to the score maps shown in Fig. 7. Since consecutive frames are often too similar to reveal meaningful inconsistencies, we compute MEt3R over a 5-frame interval (i.e., between frame $i$ and $i+5$). Our ablation study shows that adding the proposed scene loss improves multi-view consistency compared to training without it, consistent with prior findings (Alonso et al., 2025) that Chamfer loss alone does not effectively enforce cross-view agreement. In contrast, our scene loss weights each 3D point by its stereo-derived confidence, guiding the optimization toward stronger spatial alignment across views. As shown on the right side of Fig. 7, models trained with scene loss achieve better consistency scores.

## 5    Conclusion

We introduce LSU, a novel framework for 3D reconstruction that operates effectively with single-view and sparsely sampled input images. To address the challenge of extending object-level reconstruction to the scene level, we introduce a new approach for computing scene-level reconstruction loss. Additionally, depth and normal losses are incorporated to guide the Gaussian-based reconstruction process. Experimental results demonstrate the model's strong generalization and extrapolation capabilities across a wide range of environments.

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
