# OpenReview forum: "Learning to See the Unseen: Few-Shot 3D Scene Reconstruction via Diffusion and Gaussian Fields"
_TMLR — Rejected by TMLR_

### Review · Reviewer_NH1Y · 2026-04-12

**Summary Of Contributions:**

The paper introduces "Learning to See the Unseen" (LSU), a two-stage framework for 3D scene reconstruction from single or sparse views. It leverages a Scene Video Diffusion Model to generate consistent novel views with explicit geometric supervision, which are then lifted into a 3D Gaussian Splatting representation refined by depth and normal priors.

**Strengths:** The approach elegantly bridges the gap between 2D generative priors and 3D physical constraints. It offers a unified solution that handles both single-view and sparse-view scenarios well, demonstrating strong performance on RealEstate10K and impressive cross-dataset generalization on KITTI and Mip-NeRF.

**Weaknesses:** The pipeline is computationally complex, relying heavily on multiple large external models (like DUST3R and Depth Anything v2). Additionally, the generative nature of the model inevitably leads to hallucinations in entirely unobserved regions.

**Audience:**

Yes

**Audience Explanation:**

The fusion of video diffusion models with 3D Gaussian Splatting to solve sparse-view ambiguities is a highly active and relevant research direction. TMLR readers interested in 3D computer vision and neural rendering will find this methodology valuable.

**Claims And Evidence:**

Yes

**Claims Explanation:**

The claims are well-supported by thorough quantitative metrics and qualitative visualizations. The authors evaluate against a solid set of recent baselines (e.g., MVSplat, Wonderland). Furthermore, the ablation studies are convincing, effectively isolating the impact of the depth and normal priors, while the use of the MEt3R metric clearly validates that the proposed scene loss improves multi-view spatial alignment.

**Requested Changes:**

**Critical to secure acceptance:**
1. Provide a detailed computational complexity analysis. Since the pipeline uses several heavy priors, please report training/inference times and GPU memory requirements compared to primary baselines like Wonderland or MVSplat.
2. Clarify the use of text prompts in the experimental setup. How are these descriptions obtained or utilized for large-scale evaluations on datasets like RealEstate10K or KITTI?

**To simply strengthen the work:**
1. Include visual examples of failure cases (e.g., severe hallucinations or geometric distortions in unobserved areas) in the appendix to guide future research.
2. Explicitly state the assumptions regarding camera poses during inference—are extrinsics assumed to be known perfectly, or are they jointly estimated?

---

> ### Author Response · Authors · 2026-05-08
> **Comment**
>
> We thank the reviewer for their insightful comments and suggestions, and provide our responses to them below.
>
> **Requested Changes**
>
> **Critical to secure acceptance:**
>
> 1. Cat3D takes about 16 minutes per scene, while ViewCrafter takes more than 6 minutes to generate a 25-frame novel-view video rather than directly producing Gaussians, and Wonderland takes around 5 minutes on an A100. Our method takes approximately 7 minutes per scene on an NVIDIA A6000 GPU with 48 GB of memory. Since we do not have access to an A100 machine for experimentation, and the reported runtimes are obtained on different hardware setups, we believe a direct speed comparison would not be entirely fair. In addition, controlled benchmarking on the same A6000 setup is not possible for some baselines, such as Wonderland and Cat3D, since their code is not publicly available. While MVSplat is much faster, we do not consider it for comparison because it is not a generative method; against other generative approaches, our latency is in a similar range
>
>     Training complexity is not directly comparable across baselines because it is reported in different forms: Wonderland uses a progressive 200K + 100K training schedule, ViewCrafter uses a two-stage 50K + 5K schedule, and MVSplat is trained for 300K iterations on a single A100 GPU. Our training takes about 5 days, and more details are provided in Appendix B.2
>
> 2. As described in Appendix B.2, during training, captions are generated using Groq from a single static image randomly selected from each scene.
>
> **To simply strengthen the work:**
>
> 1. Limitations for single-image hallucination and dynamic scenes are illustrated in Figs. 6 and 7 in the updated Appendix E.
> 1. The path planning for inference is done using known extrinsic camera parameters, and it’s updated in Appendix B.2

---

> > ### Comment · Reviewer_NH1Y · 2026-05-22
> >
> > Thank you for the detailed response.
> >
> > I have checked the authors' response regarding my main concerns: computational complexity, the use of text prompts, failure cases, and camera-pose assumptions during inference.
> >
> > The clarification on the caption generation process, namely that captions are generated using Groq from a randomly selected static image during training, addresses my concern about the use of text prompts. The added discussion of known extrinsic camera parameters for inference also addresses my question about camera-pose assumptions. I also appreciate the added failure-case examples for hallucination and dynamic scenes in the appendix.
> >
> > Overall, my main requested changes have been sufficiently addressed. I do not have additional major concerns.

---

### Review · Reviewer_EcVL · 2026-04-29

**Summary Of Contributions:**

This paper addresses a critical challenge in 3D scene reconstruction: existing methods rely heavily on dense multi-view inputs, struggle with geometric ambiguity in occluded/unobserved regions, and lack a unified framework that handles both single-view and sparse-view settings. To fill this gap, the authors propose LSU (Learning to See the Unseen), an end-to-end framework that couples generative novel-view synthesis with Gaussian-based 3D scene reconstruction.

The core contributions of the work are summarized as follows:

- A unified framework for high-fidelity 3D scene reconstruction that works seamlessly with both single-image and sparse-view inputs, addressing a key limitation of prior methods that are tailored to only one input setting.

- A Scene Diffusion Module (SDM) that conditions on sparse views and optional text prompts to synthesize geometrically consistent novel views, with a novel scene-level geometric supervision strategy that constrains the diffusion process via 3D structural consistency to improve cross-view spatial alignment.

- A geometry-aware Gaussian reconstruction module that leverages depth and surface normal priors to refine 3D Gaussian primitives, boosting geometric accuracy, background coherence, and rendering fidelity for both observed and occluded regions.

- Extensive experiments demonstrating that LSU achieves state-of-the-art (SOTA) performance on the RealEstate10K dataset, with strong cross-domain generalization to unseen datasets including KITTI and Mip-NeRF, preserving fine-grained details while recovering accurate global scene geometry.

**Audience:**

Yes

**Audience Explanation:**

The findings of this paper will be of significant interest to multiple large subsets of TMLR’s audience, which focuses on machine learning for intelligent systems, including core computer vision, 3D vision, generative modeling, and robotics research communities.

First, the paper addresses a fundamental, long-standing challenge in 3D scene understanding: high-fidelity geometry recovery from sparse or single-view inputs. This is a core research direction for the 3D vision and graphics community, and the proposed unified framework for single-view and sparse-view reconstruction fills a notable gap in existing literature. Researchers working on generalizable 3D reconstruction, novel view synthesis, and 3D Gaussian Splatting will be highly interested in the technical innovations and empirical results presented.

Second, the work advances the state of generative modeling for 3D vision, a rapidly growing area within the machine learning community. The proposed scene-level geometric supervision strategy provides a principled way to enforce 3D structural consistency in diffusion-based novel view synthesis, a design that can be extended to a wide range of other 3D generation tasks. This will be relevant to researchers working on diffusion models, multi-modal learning, and generative 3D vision.

**Broader Impact Concerns:**

N/A.

**Claims And Evidence:**

No

**Claims Explanation:**

There is a complete lack of analysis of the method’s computational efficiency (inference latency, memory footprint, rendering speed), a core metric for 3D Gaussian-based methods that are widely adopted for their efficient rendering.

Ablation studies for key technical components (the confidence-weighted mechanism in the scene loss, the Cross-Modal Fusion Block) are missing, leaving the contribution of these designs unvalidated.

Limitations of the method (dynamic scenes, large-scale scenes, object hallucination in single-view settings) are only briefly mentioned, with no experimental or quantitative analysis to characterize failure modes.

Implementation details for several key modules are overly brief, which may hinder reproducibility of the work.

**Requested Changes:**

- The paper centers on 3D Gaussian Splatting, a method widely adopted for its industry-leading rendering efficiency, yet provides no quantitative analysis of LSU’s computational cost. To validate the practical utility of the framework, you must add a direct comparison of inference time per scene, GPU memory usage, and rendering speed (FPS) against key baselines (e.g., pixelSplat, MVSplat, Wonderland) on a consistent hardware setup. This is essential to support the claim that LSU is a viable solution for real-world 3D reconstruction tasks.

- The current ablation validation is incomplete for two core proposed designs:
The confidence-weighted mechanism in the scene-level loss: You claim that confidence-based filtering and weighting improve supervision robustness, but provide no ablation comparing performance with and without this strategy (e.g., uniform weighting vs. confidence weighting).
The Cross-Modal Fusion Block (CMFB): You state that CMFB enables bidirectional visual-textual fusion to improve novel view synthesis, yet no ablation is provided to verify the module’s contribution to final reconstruction performance.
You must add these ablation experiments to confirm that these components drive the observed performance gains, rather than confounding factors.

- You only briefly mention dynamic scenes, large-scale scenes, and single-view object hallucination as limitations in the conclusion, with no supporting experimental analysis. To meet TMLR’s standards for transparent and rigorous research, you must:
Add at least qualitative results (and quantitative metrics if possible) demonstrating LSU’s performance on dynamic scenes (e.g., a subset of RealEstate10K with moving objects) and extremely large-scale scenes, with clear characterization of failure modes.
Add a quantitative evaluation of object hallucination in the single-view setting (e.g., the rate of hallucinated objects not present in the ground-truth scene) to transparently present the tradeoffs of the generative design.

- In Table 4, you report that LSU outperforms MVGD (a SOTA 2-view reconstruction method) on the RE10K Interpolation benchmark, yet provide no qualitative visual comparison. To fully validate LSU’s superiority over this strong baseline, you must add qualitative results comparing reconstruction details, geometric consistency, and occlusion handling between LSU and MVGD for 2-view input settings.

---

> ### Author Response · Authors · 2026-05-08
> **Comment**
>
> **Requested Changes**
>
> - Latency: Cat3D takes about 16 minutes per scene, while ViewCrafter takes more than 6 minutes to generate a 25-frame novel-view video rather than directly producing Gaussians, and Wonderland takes around 5 minutes on an A100. Our method takes approximately 7 minutes per scene on an NVIDIA A6000 GPU with 48 GB of memory. Since we do not have access to an A100 machine for experimentation, and the reported runtimes are obtained on different hardware setups, we believe a direct speed comparison would not be entirely fair. In addition, controlled benchmarking on the same A6000 setup is not possible for some baselines, such as Wonderland and Cat3D, since their code is not publicly available. While MVSplat is much faster, we do not consider it for comparison because it is not a generative method; against other generative approaches, our latency is in a similar range
>
>     Training complexity is not directly comparable across baselines because it is reported in different forms: Wonderland uses a progressive 200K + 100K training schedule, ViewCrafter uses a two-stage 50K + 5K schedule, and MVSplat is trained for 300K iterations on a single A100 GPU. Our training takes about 5 days, and more details are provided in Appendix B.2
>
> - We would like to clarify that Appendix D already includes the relevant ablations, with Table 1 covering CMFB and text and Table 2 covering scene loss, and we further provide an additional comparison between uniform and confidence-weighted scene loss here.
>
> | Method | PSNR $\uparrow$ |
> | --- | --- |
> | w/o scene loss | 25.8 |
> | Uniform scene loss | 26.1 |
> | Confidence-weighted scene loss | 26.5 |
> - Limitations for single-image hallucination and dynamic scenes are illustrated in Figs. 6 and 7 in the updated Appendix E.
> - The qualitative comparison for MVGD has been updated in Figure 5, and details of the comparison have also been added to the discussion section in Appendix C.

---

### Review · Reviewer_JfvG · 2026-05-01

**Summary Of Contributions:**

LSU is a two-stage pipeline method for few-shot 3D scene reconstruction. Stage 1 is a video diffusion module (Scene Diffusion Module, SDM) conditioned on DUSt3R point cloud renders and a CLIP text embedding, fused through a Cross-Modal Fusion Block (CMFB). A confidence-weighted Chamfer loss supervises geometry during finetuning of the diffusion model. Stage 2 is a feed-forward Gaussian head supervised by Depth Anything depth and normal losses derived from the depth maps. Evaluation includes RealEstate10K (in-domain), KITTI (cross-dataset, quantitative), Mip-NeRF (qualitative), and few-shot interpolation on RealEstate10K and Mip-NeRF.

**Audience:**

Yes

**Audience Explanation:**

Sparse and single-image 3D reconstruction with diffusion priors is an active area of generative 3D and novel-view synthesis.

**Claims And Evidence:**

No

**Claims Explanation:**

Claims lack adequate support.

- The RealEstate10K numbers do not reconcile across tables. LSU appears at 25.34 PSNR in Tab 1 (single-target view, 30-frame window), 21.56 in Tab 2 (multi-target view), 23.10 in Tab 3 (camera-guided), 28.61 in Tab 4 (2-view few-shot), and 26.1–26.5 in Supp Tab 1 and Supp Tab 2. The supplementary numbers are in between the main paper numbers, but do not match any single protocol. The configuration behind the 26.5 ablation result is also not specified.

- The scene loss is posed as a primary contribution, but only shows a 0.7 PSNR gain over baseline. Also, Tab 2 in the supplementary doesn't really tally with the ablation numbers in Tab 5 in the main paper.

- The camera-guided exp in Tab 3 supposedly conditions LSU's diffusion model on the gt camera poses. But it's not clear whether the baselines ViewCrafter and Wonderland are also given gt camera poses. If not, the comparison is unfair.

- The cross-dataset generalization claim is only backed by 1 dataset (KITTI). MipNeRF scenes appear only qualitatively in Fig 4. Also, cross-dataset performance on PSNR and LPIPS is identical with Flash3D.

**Requested Changes:**

Address issues cited in the claims section.

In addition, the method is not positioned well wrt prior work. For example, ViewCrafter has a similar algorithm - Dust3r init, conditioning of a video diffusion model on point cloud renders, iterative refinement of the point cloud and final 3dgs optimization. I believe the scene loss is at least in part inspired from ReconX (Liu et al., 2024). The depth and surface normal loss formulations have previously been seen in MultiDiff (Muller et al., CVPR'24), GenWarp (Seo et al., Neurips'24), DN-Splatter (Turkulainen et al., WACV'25). If there is additional novelty over these papers, it should be stated more clearly and supported with experiments.

---

> ### Author Response · Authors · 2026-05-08
> **Comment**
>
> **Claims lack adequate support.**
>
> - The configuration for Table 1 and Table 2 in the Appendix follows the same procedure as Table 1 in the main paper. The difference in PSNR values in Tables 1 and 2 of the Appendix and in Table 1 of the main paper arises because the Appendix results correspond to outputs after Stage 1, whereas the main paper reports results after Stage 2. We report Stage 1 results in the ablation study done in the Appendix because both modules under comparison belong to Stage 1 of the baselines, and these views serve as baselines for constructing the Gaussians. As a result, the PSNR values for the novel views are higher at this stage than those obtained after rendering from the Gaussians in the subsequent stage.
> - If we compare other existing works, their complete method itself shows less than 1 dB difference in PSNR with respect to other methods thus 0.7 dB difference just with the help of scene loss seems reasonably good especially in this literature.
>
>     Table 5 in the main paper follows the experimental setup of Table 2 in the main paper, while the ablations Tab1 and Tab2 in the Appendix follow the setup of Table 1 in the main paper. Thus, both these protocols are different. For a fair comparison, we present the results of Table 5 using the same procedure as that used for the ablations in the Appendix, as follows.
>
>     | RGB | Depth | Normal | PSNR $\uparrow$ |
>     | --- | --- | --- | --- |
>     | $\checkmark$ | $\times$ | $\times$ | 24.46 |
>     | $\checkmark$ | $\checkmark$ | $\times$ | 25.01 |
>     | $\checkmark$ | $\checkmark$ | $\checkmark$ | 25.34 |
> - Wonderland, Viewcrafter, and ours are given similar ground truth poses for fair comparison.
> - We would like to clarify that Figures 3 and 4 in Appendix C include results on two additional datasets, further demonstrating cross-dataset generalization.
>
>     Since prior work reports only qualitative results on the MipNeRF dataset, we follow the same evaluation protocol for consistency. Moreover, the implementation details and code for these methods are not publicly available for quantitative analysis with other datasets.
>
>     We accept the fact the there is not much difference in the values compared to Flash3d for cross dataset but it should also be considered that Flash3D is not a generalizable framework and works only for a single image and cannot go beyond a small angle from the given view. These facts weigh Flash3D down massively.

---

> > ### Author Response · Authors · 2026-05-08
> > **Comment**
> >
> > We thank the reviewer for highlighting the need for clearer positioning.
> >
> > **Difference from ViewCrafter.**
> >
> > ViewCrafter is primarily a **novel view synthesis** framework, where 3D Gaussian Splatting (3DGS) is used only as an application via standard optimization. In their pipeline, Gaussians are **optimized per scene**, requiring test-time optimization for each new input. In contrast, our method addresses **3D reconstruction directly** by learning a **feed-forward Gaussian prediction module**, which estimates Gaussian parameters in a single forward pass without per-scene retraining or optimization. This is a fundamental methodological difference: we replace optimization-based 3DGS with a **generalizable reconstruction network**.
> >
> > Further, while both methods may use DUSt3R initialization, this is not unique to ViewCrafter - many recent works adopt DUSt3R as a strong geometric prior. We follow this standard practice purely for initialization.
> >
> > More importantly, our approach introduces two components absent in ViewCrafter: Firstly, a **scene-level geometric loss** that directly supervises the diffusion process via 3D consistency. Secondly, a **cross-modal text–visual guidance mechanism (CMFB)** that improves semantic and geometric coherence during generation.
> >
> > The effectiveness of the scene loss is validated through ablations and improved multi-view consistency (Fig. 7, MET3R), demonstrating its contribution beyond standard diffusion-based synthesis.
> >
> > **Difference from ReconX .**
> >
> > ReconX uses 3D features as **conditioning inputs** to diffusion. In contrast, our method introduces an explicit **scene-level geometric loss that operates in 3D space and backpropagates into the diffusion model**. Thus, rather than conditioning on 3D representations, we **enforce geometric consistency as a training objective**, which is a fundamentally different use of 3D information. Fig 5 in the updated appendix C shows a clear difference in the quality of the obtained results.
> >
> > **Difference from MultiDiff and GenWarp.**
> >
> > MultiDiff and GenWarp incorporate depth as a **conditioning signal within the diffusion model**, where depth from a single input view is injected to guide novel view synthesis.
> >
> > In contrast, our method uses depth for a **fundamentally different objective and in a different component of the pipeline**. Specifically, depth is not used to condition the diffusion process; instead, it is used within the **Gaussian reconstruction module** to supervise and anchor the prediction of 3D Gaussian parameters.
> >
> > Thus, while prior works leverage depth to guide **image generation**, we use depth to enable **structured 3D reconstruction** via a feed-forward Gaussian prediction network. This separation of roles allows our method to maintain stable diffusion training while enforcing geometric consistency during reconstruction.
> >
> > **Difference from DN-Splatter.**
> >
> > DN-Splatter extends standard 3DGS by incorporating **depth and normal losses during optimization**. In contrast, our method uses depth and normal priors within a **feed-forward Gaussian reconstruction network**, avoiding per-scene optimization entirely.

---

### Comment · Action_Editor_fVqu · 2026-05-22
**Please check the authors' response and engage**

Dear reviewers,

Thank you for your contributions to the review of this submission.

Since the authors have submitted their responses to your review comments, please take a look at the responses and see if they addressed your concerns.

It would be great if you could reply to the authors to let them know whether your concerns were addressed and whether there are any concerns outstanding.

Thanks,
AE

---

### Decision · Action_Editor_fVqu · 2026-06-17

**Recommendation:** Reject

**Audience:**

Yes

**Audience Explanation:**

The paper addressed an important problem with an interesting solution about 3D reconstruction, which would be of interest to at least some individuals in TMLR's audience.

**Claims And Evidence:**

No

**Claims Explanation:**

This submission received review comments from three expert reviewers. The authors provided responses to the comments, after which there were a few rounds of discussions, either between the authors and the reviewer, or between the AE-Reviewers. From the discussion, some of the concerns initially raised by the reviewers were addressed by the authors, and one reviewer recommended *Accept*, while the other two reviewers still *Leaning Reject*. The remaining concerns are mainly around the evidence to support the claims made, including the inconsistency among the experimental results across several places (e.g. the main paper, the rebuttal, and the appendix); the inconsistency between the performance and the claims regarding the two stages; and the claim of strong cross-dataset generalization. In the end, two out of three reviewers agreed on the lack of evidence to support the claims made in the submission.

---

> ### Author Response · Authors · 2026-06-20
> **Request for Reconsideration of Decision**
>
> Dear Action Editor,
>
> Thank you for your time and for handling our submission. We appreciate the efforts of the reviewers and the editorial team throughout the review process.
>
> After carefully reading the decision letter, we would like to respectfully request reconsideration of the decision, or alternatively, the opportunity to submit a revised version addressing the remaining concerns.
>
> Our primary concern is that the final decision cites issues that, from our understanding, had already been addressed in the rebuttal and revision materials, yet there was no subsequent discussion with the two reviewers who maintained their concerns despite multiple requests from us for further reviewer engagement.
>
> Specifically:
>
> 1. **Reviewer 1's concern regarding inconsistencies in the ablation results**
>
>    The reviewer raised concerns about inconsistencies between results reported in different parts of the submission. In our rebuttal, we explicitly clarified that the results presented in the main paper correspond to the complete two-stage pipeline, whereas the ablation results reported in the supplementary material were intentionally conducted on Stage 1 alone and therefore did not include the effects of Stage 2. This distinction was clearly stated in our response and explains the apparent discrepancy. Unfortunately, we did not receive any follow-up discussion from the reviewer indicating whether this clarification had been seen or whether additional evidence was required.
>
> 2. **Reviewer 2's concern regarding missing ablations**
>
>    The reviewer requested several ablation studies that were already included in the supplementary material submitted with the paper. Our rebuttal pointed to the relevant sections and tables containing these experiments. However, the final decision still cites insufficient evidence, suggesting that these supplementary analyses may not have been fully considered. Again, there was no subsequent interaction from the reviewer after our clarification.
>
> We would also like to note that, despite multiple emails sent to the Action Editor during the discussion period, we did not receive any engagement from these reviewers following our rebuttal. As a result, we were unable to determine whether our clarifications were insufficient, whether additional experiments were expected, or whether the reviewers had simply not had an opportunity to examine the updated material.
>
> Given that the final decision relies substantially on concerns that we believe were either clarified or already addressed through existing experimental evidence, we respectfully feel that the manuscript may not have received a complete evaluation of the rebuttal and supplementary materials.
>
> We therefore kindly request consideration of one of the following options:
>
> * Reopening the review for a limited discussion period so that the remaining concerns can be directly addressed with the reviewers; or
> * Allowing submission of a minor revision focused on improving the presentation, clarifying the distinction between the stage-wise and full-pipeline results, and consolidating the experimental evidence into clearer tables to avoid any ambiguity.
>
> We fully understand that acceptance is not guaranteed and respect the editorial process. Our request is simply motivated by the belief that the remaining concerns largely arise from misunderstandings or overlooked clarifications rather than unresolved technical deficiencies.